# Acceptance of COVID-19 vaccines in sub-Saharan Africa: evidence from six national phone surveys

Shelton Kanyanda, Yannick Markhof  , Philip Wollburg, Alberto Zezza

Development Data Group, World Bank Group, Washington, DC, USA

**Correspondence to**
Dr Alberto Zezza;
azezza@worldbank.org

## ABSTRACT

**Objectives** To estimate the willingness to accept a COVID-19 vaccine in six sub-Saharan African countries and identify differences in acceptance across countries and population groups.

**Design** Cross-country comparable, descriptive study based on a longitudinal survey.

**Setting** Six national surveys from countries representing 38% of the sub-Saharan African population (Burkina Faso, Ethiopia, Malawi, Mali, Nigeria and Uganda).

**Participants** Respondents of national high-frequency phone surveys, aged 15 years and older, drawn from a nationally representative sample of households.

**Main outcome measures** Willingness to get vaccinated against COVID-19 if an approved vaccine is provided now and for free, disaggregated by demographic attributes and socioeconomic factors obtained from national household surveys. Correlates of and reasons for vaccine hesitancy.

**Results** Acceptance rates in the six sub-Saharan African countries studied are generally high, with at least four in five people willing to be vaccinated in all but one country. Vaccine acceptance ranges from nearly universal in Ethiopia (97.9%, 95% CI 97.2% to 98.6%) to below what would likely be required for herd immunity in Mali (64.5%, 95% CI 61.3% to 67.8%). We find little evidence for systematic differences in vaccine hesitancy by sex or age but some clusters of hesitancy in urban areas, among the better educated, and in richer households. Safety concerns about the vaccine in general and its side effects specifically emerge as the primary reservations toward a COVID-19 vaccine across countries.

**Conclusions** Our findings suggest that inadequate demand is unlikely to represent the key bottleneck to reaching high COVID-19 vaccine coverage in sub-Saharan Africa. To turn intent into effective demand, targeted information, sensitisation and engagement campaigns bolstering confidence in the safety of approved vaccines and reducing concerns about side effects will be crucial to safeguard the swift progression of vaccine rollout in one of the world's poorest regions.

## INTRODUCTION

As vaccination campaigns in high-income countries are accelerating, large swathes of the global population living in low-income and middle-income countries remain severely exposed to the COVID-19.[1] Sub-Saharan Africa (SSA) is home to 433 million people

## Strengths and limitations of this study

► Much of the current debate on vaccination campaigns in low-income countries in general and sub-Saharan Africa (SSA) in particular focuses on supply chain and financial factors, yet there is a dearth of robust, comparable evidence on COVID-19 vaccine hesitancy in these countries.

► We use cross-country comparable data from six national high-frequency phone surveys in SSA to fill this crucial knowledge gap.

► Our sample is drawn from large, nationally representative sampling frames that provide a rich set of demographic and socioeconomic characteristics by which we disaggregate our analysis.

► Through a set of recalibrated survey weights, our analysis adjusts for the selection biases common in other remote surveys, yet some bias may persist.

► Attitudes towards vaccines are malleable and may change over time.

living below the international absolute poverty line (about two-thirds of the global poor population).[2] For these populations, non-pharmaceutical interventions to curb the spread of the disease impinged on already precarious livelihoods.[3] Furthermore, the region is characterised by high prevalence of comorbidities and public health systems that are ill equipped to stave off the burden of mass infection.[1 3 4] Reaching large-scale vaccination coverage in SSA is thus pivotal in the global effort to halt the spread of the disease and limit its toll on lives and livelihoods.[5–7]

Recent calls to action by international stakeholders and a growing body of scholarly research focus on supply chain and financial factors to safeguard sufficient vaccine availability in SSA. The COVAX initiative aims to provide doses sufficient to vaccinate up to 20% of the population in the region, far below the target population required for a coverage of 60% or 321.5 million individuals.[8 9] Still, this would leave an estimated financing gap of over $10 billion in SSA.[10] A June 2021 resolution from the G7 pledging

1 billion COVID-19 vaccine doses for low-income countries offers hope for those supply gaps being filled. While these considerations concern the availability of vaccines, another key factor for the success of vaccination campaigns in SSA is the willingness to be vaccinated within the population. Yet, there is a dearth of large-scale evidence on vaccine acceptance in low-income countries in general and SSA in particular where previous studies came to diverging conclusions.[11–14]

Based on a cross-country comparable sample with national scope from six sub-Saharan African countries, we fill this knowledge gap by providing estimates of vaccine acceptance for a population representative of around 416 million people, 38% of the population of SSA.[15] Drawing on the high-frequency phone surveys (HFPS) based on prepandemic sampling frames from nationally representative, face-to-face (FtF) household surveys supported by the World Bank's Living Standards Measurement Study – Integrated Survey on Agriculture (LSMS-ISA), we are able to link COVID-19 vaccine acceptance rates to a rich set of demographic and socioeconomic characteristics. Along with recalibrated sampling weights, this allows our study to provide robust insights into the likelihood of the current efforts to ensure sufficient supply of vaccination doses to also meet adequate demand in SSA, identify clusters of hesitancy and contribute to a swift rollout of vaccination campaigns where they will be needed most.

## METHODS
### Data and survey instrument
We use data from the national longitudinal HFPS on COVID-19 conducted in Burkina Faso, Ethiopia, Malawi, Mali, Nigeria and Uganda. Survey implementation was led by the respective national statistical agencies—the Burkina Faso National Institute of Statistics and Demography; Laterite Ethiopia in collaboration with the Central Statistical Agency of Ethiopia; Malawi National Statistical Office; Mali National Institute of Statistics; Nigeria Bureau of Statistics; and Uganda Bureau of Statistics—and supported by the World Bank Living Standards Measurement Study and the Poverty and Equity Global Practice. The HFPS on COVID-19 have been implemented monthly since May 2020, aiming to gauge the impact of the COVID-19 pandemic on individual and household attitudes, socioeconomic and health outcomes.[16] Section 2 of the online supplemental appendix 1 includes more detail on the conception and implementation of the HFPS.

The HFPS rounds in which vaccine hesitancy was covered in a cross-country comparable fashion were conducted in September 2020 (Ethiopia: round 6), October–November 2020 (Malawi: round 5; Mali: round 5; and Nigeria: round 6) and December 2020 (Burkina Faso: round 5 and Uganda: round 4). The primary survey questions of interest, posed to each phone survey respondent, was: '*If an approved vaccine to prevent coronavirus was available right now at no cost, would you agree to be vaccinated?*' with response options '*yes*', '*no*' and '*not sure*'. If respondents answered '*no*' or '*not sure*', this was followed up by a question about possible reasons for refusing to be vaccinated: '*What are the reasons you would not agree to be vaccinated?*' with response options:
1. 'I don't think it will be safe'
2. 'I am worried about the side effects'.
3. 'It is against my religion'.
4. 'I am not at risk of contracting COVID-19'.
5. 'I don't think it will work'.
6. 'I am against vaccine in general'.
7. Other.

We are interested in the association between willingness to be vaccinated and a standard set of demographic and socioeconomic variables, such as gender, age, income and education. These are available for respondents either from the HFPS directly or from pre-COVID-19 FtF surveys, which served as the sampling frames for the HFPS. We further draw on multiple rounds of the HFPS to create variables on respondents' attitudes towards the COVID-19 emergency and how it has affected them and their families. These include willingness to be tested for COVID-19, rating of government response and whether households received any government assistance during the pandemic. The data as well as survey instruments and basic information documents are available publicly on the World Bank Microdata Library under the High-Frequency Phone Survey collection.[17]

### Sampling and sample representativeness
The samples for the HFPS were drawn from mobile phone numbers recorded during data collection for nationally representative FtF household surveys implemented prior to the COVID-19 pandemic with support from the World Bank LSMS-ISA programme, specifically the Burkina Faso Enquête harmonisée sur les conditions de vie des ménages 2018/19, Ethiopia Socioeconomic Survey 2018/19, Malawi Integrated Household Panel Survey 2019, Mali Enquête harmonisée sur les conditions de vie des ménages 2018/19, Nigeria General Household Survey – Panel 2018/19 and Uganda National Panel Survey 2019/20. At least one phone number was recorded for all households with access to a phone, including through a contact person outside the household, such as a friend or neighbour. For the HFPS, Ethiopia, Malawi and Uganda attempted to contact all households with available phone numbers, while Burkina Faso, Nigeria and Mali selected a random subsample of phone numbers to call (online supplemental table A1).

In the absence of universal access to a mobile phone, the HFPS households are likely to be selected samples of the nationally representative FtF survey samples of households.[18] The publicly available HFPS datasets therefore come with recalibrated household survey weights to counteract potential selection biases. The recalibration model takes advantage of the rich information on households with and without access to a phone

from the pre-COVID-19 FtF surveys[3] [19] and follows a methodology proposed in a reference methodological paper on this subject.[20] The covariates included in the logistic response propensity model included household characteristics such as household size, urban/rural residence, economic status, education, sex and age of the household head and a number of other household characteristics selected based on their observed bias in the interviewed sample. The effectiveness of this weight recalibration in overcoming selection biases has been documented in HFPS data from four of the six countries we study, such that household-level estimates based on the HFPS data can be broadly considered representative at the national level.[18]

Furthermore, the HFPS survey questions on vaccinations were asked only to the household's main respondent, who had to be 15 years or older, as it is impractical to interview all individuals in a household in a phone survey. The selection of main respondents was not randomised so that the group of respondents is likely not representative of the general population of adults at the individual level. Rather, respondents tend to be household heads or their spouses, better educated and slightly older, as a recent study documents.[21] An additional recalibration of survey weights for individual-level analysis analogous to the approach described above improves representativeness but cannot overcome respondent selection biases in all variables and increases the variance of the estimates.[21]

Considering these potential shortfalls in the representativeness of our data, we present our main results first with the standard HFPS household weights and then, as a robustness and sensitivity check for potential respondent selection biases, with the recalibrated individual-level weights. A summary of unweighted individual characteristics of FtF adults vis-à-vis HFPS adults is presented in online supplemental table A2.

### Statistical analysis

The statistical analysis proceeds in several steps. First, we estimate the weighted mean of willingness to get vaccinated by country and within countries by sex of respondent, residence (urban and rural) and income quintile, as well as reasons for vaccine hesitancy, using the recalibrated household weights. Second, we explore how individual and household characteristics, such as education and expenditure, correlate with the willingness to get vaccinated in a set of multivariate logit regressions, again using household weights. To assess how much the differences in the attributes of respondents vis-à-vis the general adult population may affect the representativeness of our results, we assess the sensitivity of the results to using individual-level recalibrated weights instead of the public use household weights (see Discussion).[21]

### Patient and public involvement

This research did not involve consultation with patients or the public.

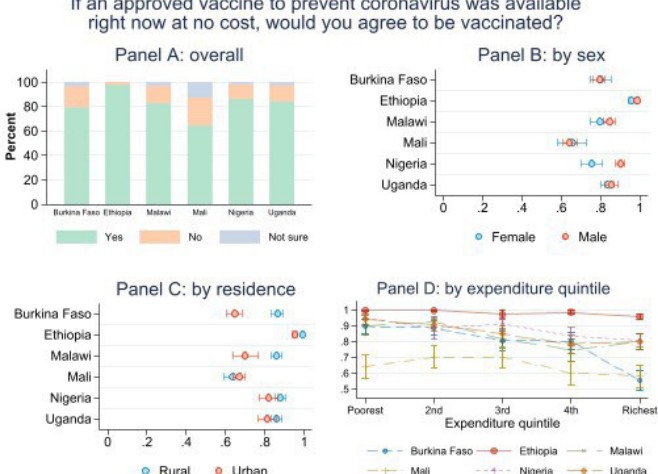

**Figure 1** Vaccine hesitancy overall, by sex, by residence and by expenditure quintile.

## RESULTS

### Descriptive results

Overall, we find high levels of willingness to be vaccinated: acceptance is estimated to be nearly universal in Ethiopia at 97.9% (95% CI 97.2% to 98.6%) and very high in Nigeria (86.2%, 83.9% to 88.5%), Uganda (84.5%, 82.2% to 86.8%), Malawi (82.7%, 80.0% to 85.4%) and Burkina Faso (79.5%, 76.9% to 82.1%). In these countries, at least four in five respondents would agree to be vaccinated if an approved vaccine was made available for free to them (figure 1, panel A). Acceptance is somewhat lower only in Mali where less than two-thirds of respondents (64.5%, 61.3% to 67.8%) reported their willingness to be vaccinated. Pairwise tests of statistical significance of the difference in hesitancy between countries are reported in online supplemental table A3. Notably, Mali is also the only country in which a non-negligible share (12.4%) is uncertain about their answer, a fact that accounts for the majority of the lower acceptance rates in Mali. For the following analysis, we focus on acceptance rates and do not distinguish between respondents rejecting to be vaccinated and those that are not sure. Pooling the data from all countries and weighting them by their respective popualtion sizes yields an overall mean acceptance rate of 87.6% (86.4% to 88.8%) across the six countries (online supplemental table A4).

In panel B of figure 1, we disaggregate acceptance rates by respondent sex. Except for Nigeria where acceptance is statistically significantly higher among male than female respondents (90.1%, 87.6% to 92.6% and 75.7%, 70.4% to 80.9%, respectively) and in Ethiopia where there is a small yet statistically significant difference (98.7%, 97.9% to 99.4% and 95.8%, 94.1% to 97.5%, respectively), we do not find answers to differ between men and women. Similarly, panel C reports differences in willingness to be vaccinated against COVID-19 between rural and urban areas with higher acceptance in rural areas in Burkina Faso, Ethiopia and Malawi at the 95% confidence level.

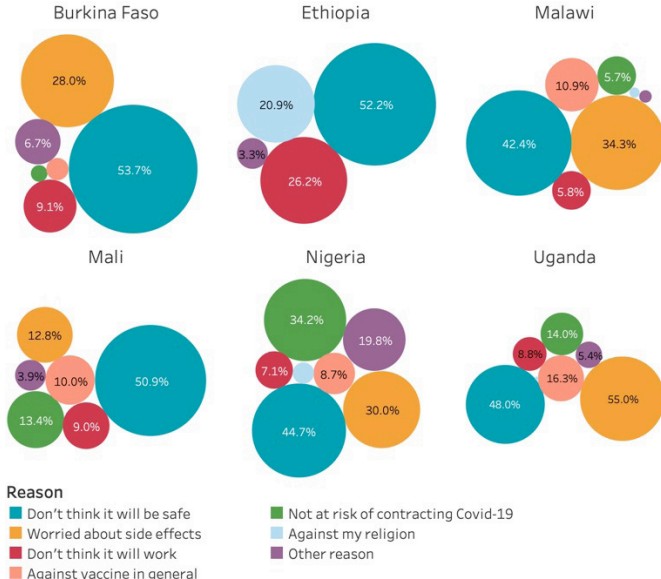

**Reason**
- Don't think it will be safe
- Worried about side effects
- Don't think it will work
- Against vaccine in general
- Not at risk of contracting Covid-19
- Against my religion
- Other reason

**Figure 2** Reasons for COVID-19 vaccine hesitancy.

One distinct feature of our data is the ability to tap into the rich pre-COVID-19 baseline data from the FtF household surveys that served as sampling frames. This way, we are able to determine respondent household's position in the national expenditure distribution and disaggregated acceptance rates by expenditure quintile (panel D). We generally find a downward sloping pattern in which acceptance is higher among poorer households and lowest among the richest households. This pattern is particularly evident in Burkina Faso but also noteworthy in Uganda and Nigeria. Conversely, acceptance is high throughout all expenditure quintiles in Ethiopia and not significantly lower for the richest two quintiles in Mali.

For those respondents who were hesitant to be vaccinated, the survey asked to provide a reason. From figure 2, it is evident that safety concerns were paramount ranging from 53.7% (36.1% to 71.3%) of those reporting they would refuse to be vaccinated in Burkina Faso to 42.4% (33.8% to 50.9%) in Malawi, despite the wording of the questions making it explicit that the vaccine would be officially approved. Other notable reasons include worries about the potential side effects of the vaccine (between 55% (46.9% to 63.0%) in Uganda and 12.8% (9.1% to 16.5%) in Mali) and the belief not to be at risk of contracting the virus (34.2% (25.2% to 43.3%) in Nigeria to 0.1% (0% to 2.1%) in Burkina Faso).

## Correlates of hesitancy

Exploiting the richness of our data, we can explore correlational patterns between willingness to be vaccinated for COVID-19 and a large set of respondent and household characteristics. Table 1 confirms in a multivariate setting the patterns observed graphically earlier. In general, men tend to be more willing to take the vaccine, though we only detect a statistically significant effect in Nigeria. Respondents in urban areas are more sceptical

with significant differences in Burkina Faso, Ethiopia and Malawi.

An association we observe across countries is between education and vaccine hesitancy. In Burkina Faso, Ethiopia, Malawi and Nigeria, those with more years of education are significantly less willing to be vaccinated while coefficient signs point in the same direction in Mali and Uganda but are not significant. Lastly, we do not find a pattern according to respondents' age and a mixed picture according to the role of the respondent in the household. Here, household heads that constitute the majority of our sample (see online supplemental table A2) are more willing to be vaccinated in Ethiopia, more hesitant in Mali and not significantly different from other household members in the remaining four cases.

A pattern we observe in several countries is a higher hesitancy towards the vaccine in richer households compared with the poorest expenditure quintile, with coefficients significant in Mali and Uganda for the richest two quintiles, for the top quintile in Burkina Faso and Nigeria and the third and fourth quintiles in Malawi.

The data further show a strong association between vaccine hesitancy and the willingness to be tested for COVID-19 suggesting similar underlying reasons for (or against) testing and getting vaccinated. Another hypothesis we explore is whether there is an association between the receipt of some assistance during the pandemic, for example, through one of the large-scale social protection programmes launched to combat the fallout of the pandemic and willingness to receive a free, approved vaccine against COVID-19. While coefficient signs mostly point to a positive association, the effect is not statistically significantly different from zero despite all countries in our sample launching some social protection response to the crisis.[22]

A last hypothesis we pursue concerns the relationship between trust in and satisfaction with the government and willingness to be vaccinated. For example, more sceptical individuals towards the government or those dissatisfied with its crisis management may be more reluctant to accept a state-provided vaccine.[12 23] We only have data on trust in the government's crisis management in Malawi and satisfaction in Mali and Nigeria, but find our hypotheses are confirmed in bivariate logistic regressions (online supplemental table A5). However, after controlling for all other factors in table 1, the coefficients are no longer significant.

## DISCUSSION
### Principal findings
Our study uses cross-country comparable data from the national longitudinal HFPSs on COVID-19 in six sub-Saharan African countries to estimate acceptance rates for an approved, free COVID-19 vaccine. By linking phone survey data to the nationally representative, large-scale, FtF household surveys that served as sampling frames for the HFPS, we provide robust estimates of the

**Table 1** Correlates of hesitancy

Correlates of hesitancy – marginal effects from multivariate logit regressions

| Variables | (1) Burkina Faso | (2) Ethiopia | (3) Malawi | (4) Mali | (5) Nigeria | (6) Uganda |
|---|---|---|---|---|---|---|
| Respondent is male | −0.0388 | 0.00913 | 0.0419 | 0.0522 | 0.0932*** | 0.0179 |
| | (0.0340) | (0.00569) | (0.0311) | (0.0413) | (0.0268) | (0.0224) |
| Lives in urban area | −0.0878*** | −0.0191** | −0.0880*** | 0.0237 | −0.0151 | 0.0168 |
| | (0.0294) | (0.00858) | (0.0289) | (0.0286) | (0.0222) | (0.0212) |
| Years of education | −0.00610** | −0.00137* | −0.0135*** | −0.00452 | −0.00680** | −0.00164 |
| | (0.00252) | (0.000734) | (0.00330) | (0.00288) | (0.00286) | (0.00283) |
| Age group 30–60 years | −0.0166 | 0.00494 | 0.0152 | 0.0297 | 0.0436 | −0.0274 |
| | (0.0365) | (0.00838) | (0.0312) | (0.0458) | (0.0366) | (0.0269) |
| Age group 60+ years | 0.0446 | 0.00700 | −0.0140 | −0.0273 | 0.0499 | 0.0179 |
| | (0.0435) | (0.0141) | (0.0505) | (0.0584) | (0.0461) | (0.0341) |
| Household head | 0.0558 | 0.0171** | −0.00894 | −0.0886** | −0.0266 | −0.00563 |
| | (0.0376) | (0.00776) | (0.0381) | (0.0450) | (0.0325) | (0.0242) |
| Household size | 0.00310 | 0.000705 | 0.0103* | −0.00536 | 0.00286 | 0.00631 |
| | (0.00378) | (0.00152) | (0.00557) | (0.00358) | (0.00580) | (0.00476) |
| Expenditure, second quintile | 0.0230 | 0.00677 | 0.0198 | 0.0307 | −0.0677 | −0.0302 |
| | (0.0426) | (0.0102) | (0.0437) | (0.0441) | (0.0564) | (0.0319) |
| Expenditure, third quintile | −0.0568 | −0.0216 | −0.0953* | 0.0166 | 0.00195 | −0.0500 |
| | (0.0430) | (0.0195) | (0.0516) | (0.0431) | (0.0427) | (0.0308) |
| Expenditure, fourth quintile | −0.0357 | −0.00381 | −0.116** | −0.0932* | −0.0623 | −0.107*** |
| | (0.0442) | (0.0103) | (0.0473) | (0.0501) | (0.0458) | (0.0340) |
| Expenditure, fifth (richest) quintile | −0.164*** | −0.0111 | −0.0106 | −0.135** | −0.0696* | −0.0754** |
| | (0.0534) | (0.0108) | (0.0457) | (0.0536) | (0.0421) | (0.0346) |
| Willing to be tested for COVID-19 | 0.208*** | 0.0619*** | 0.253*** | 0.461*** | 0.221*** | 0.340*** |
| | (0.0313) | (0.00889) | (0.0389) | (0.0172) | (0.0221) | (0.0254) |
| HH received assistance during COVID-19 | 0.0490 | 0.00607 | −0.00457 | −0.00728 | 0.0152 | 0.0307 |
| | (0.0470) | (0.00886) | (0.0322) | (0.0618) | (0.0220) | (0.0194) |
| Government not trustworthy | | | −0.0455 | | | |
| | | | (0.0291) | | | |
| Satisfied with government response | | | | 0.0354 | 0.0279 | |
| | | | | (0.0450) | (0.0202) | |
| Observations | 1742 | 2654 | 1542 | 1591 | 1703 | 2106 |
| Pseudo R$^2$ | 0.183 | 0.308 | 0.137 | 0.265 | 0.218 | 0.233 |

Robust SEs in parentheses. ***p<0.01, **p<0.05, *p<0.1. Weighted logit regressions with willingness to take a free, approved vaccine as dependent variable.

willingness to be vaccinated against COVID-19 across a diverse set of demographics and respondent and household characteristics. Our headline results indicate high acceptance rates with at least four in five respondents signalling their willingness to be vaccinated in all but one of the countries studied. There is cross-country variation, with willingness to be vaccinated ranging from 64.5% in Mali (61.3% to 67.8%), where a further 12.4% are yet undecided, to near universal acceptance in Ethiopia (97.9%, 97.2% to 98.6%). We also find little evidence for systematic differences in vaccine hesitancy across gender or age but some notable clusters of hesitancy in urban areas, among those better educated and in richer households. Across countries, safety concerns about the vaccine in general and its side effects specifically emerged as the primary reservations towards a COVID-19 vaccine.

### Strengths and comparison with other studies
In relation to the existing literature, our study has several advantages in the domains of coverage and sample

selection, richness of the data and analytical depth. In terms of coverage, our study focuses on a region with yet scant evidence on willingness to be vaccinated against COVID-19. Importantly, the data we assemble are cross-country comparable and without exception based on large, nationally representative sampling frames from pre-COVID-19, FtF household surveys. This allows us to calibrate survey weights that adjust for the coverage and non-response biases that plague other studies at a time when regular FtF data collection nearly came to a complete halt.[24–26] Furthermore, our study is unique in that the pre-existing survey data from which our samples are drawn allow us to tap into a rich set of baseline individual and household characteristics. This facilitates a more rigorous analysis and disaggregation of acceptance rates, including, for instance, households' position in the national (precrisis) expenditure distribution, than has been possible previously. Our study thus stands out as a fully cross-country comparable, multivariate and inferential analysis of vaccine hesitancy on the African continent.

Compared with our study, previous studies that analyse COVID-19 vaccine acceptance rates and the reasons for refusal predominantly: (1) focus on middle-income and high-income countries, (2) represent single-country or not cross-country comparable samples, (3) study particular subpopulations such as university students, healthcare workers or participants of unrelated pre-COVID-19 studies, (4) rely on non-random sampling or (5) can only draw on a small set of demographics and characteristics to disaggregate their analysis.[14 21 27–37]

A cross-country study of 19 countries with samples obtained from commercial online panel providers found 71.5% of respondents willing to be vaccinated against COVID-19 with a rate of 65.2% in Nigeria as the only low-income country.[38] Similarly, a literature review of 31 studies covering 33 countries found COVID-19 vaccine acceptance rates at or above 70% among studies focusing on the general population but also large regional and intraregional differences and a dearth of evidence particularly from SSA.[39]

Few studies explicitly focus on acceptance rates in the poorest countries. Assembling an amalgamation of data samples with different sources, sampling methodologies and coverage, one study finds generally high acceptance rates in 10 LMICs in Asia, Africa and South America.[13] Among countries also covered in our study, acceptance rates are lower than what we find in Burkina Faso (66.5%, national phone sample obtained by random digit dialling) and Nigeria (76.2%, random sample of residents of one state from telephone list), close in urban Uganda (76.5%, random sample of households in Kampala) and very similar in rural Uganda (85.8%, non-random sample of women in 13 districts).[13] A second study in nine LMICs (including DR Congo, Benin, Malawi, Mali and Uganda from SSA) draws on a convenience sample recruited through social media and other online channels.[14] The acceptance rates found heavily depend on the hypothesised vaccine effectiveness but are generally low

(below 75%) in all countries from SSA except Uganda. Sample sizes are roughly 5% of ours. Furthermore, the ability to correct for various sample selection biases is likely limited in these studies in the absence of baseline nationally representative sampling frames. Another study in 15 African countries, including Burkina Faso, Malawi, Nigeria, Uganda (all FtF interviewing) and Ethiopia (telephone survey) found estimates for the countries also covered in our study mostly in the vicinity of our weighted figures (Ethiopia 94%, Uganda 87.5%, Burkina Faso 86%, Malawi 82.7% and Nigeria 76%).[11 40] Notably, average sample sizes are roughly half of our study's, and there is a lack of analytical detail impeding a robust inference of cross-country and cross-demographic findings. Lastly, our results contrast with a recent Afrobarometer study based on in-person interviews in five West African countries, none of which are also included in our sample. In relation to our study, the study's analysis is purely descriptive and reports low acceptance rates of 40% on average.[12] Similar to what we find, the study cites trust in vaccine safety as the key driver of hesitancy, a result that is further corroborated in the literature.[12–14 30 38 40] Furthermore, the small systematic differences we observe according to gender and age are in line with most previous findings in low-income countries as is a tendency for higher acceptance in rural areas.[12 13 38 40] As with overall acceptance rates, our finding of higher hesitancy among the more educated is in line with two of the studies covering SSA,[11 13] while another study finds mixed evidence.[12] No other study we are aware of across LMICs (and in fact few across countries of any income classification) assesses vaccine acceptance according to economic status in a manner comparable with the expenditure data we can access from the pre-COVID-19 sampling frames.

## Limitations of this study

This study uses data from HFPS with national coverage along with sampling weights specifically recalibrated for the phone surveys to be nationally representative. The weights were shown to be effective at achieving nationally representative estimates at the household level.[18] However, willingness to be vaccinated is primarily not a household-level attribute but an individual decision, though it is reasonable to expect considerable intrahousehold correlation in attitudes towards vaccination. Phone survey respondents in our data are not specifically selected to be representative of all individuals and that may limit the population-level representativeness of the results we report. Recalibration of survey weights at the individual level can partially but not fully address this concern.[21] To gauge how sensitive our results are to respondent selection biases in individual-level data, we compare the estimates with household-level weights to the same estimates with individual-level weights. In this test, large deviations between those two sets of estimates would indicate that selection biases affect our results in a fundamental way. However, we find only limited change in the estimates, regardless of which weight is used: in over three-quarters

of the cases, deviations do not exceed two percentage points, including the headline findings on willingness to be vaccinated in all study countries, and we find only three instances where the differences exceed five points (variable 'Q5' 5.8% in Burkina Faso; 'Q4' 6.2% in Mali; 'Q3' 6.0% in Nigeria; online supplemental table A6). The results from the multivariate logit regression model are also robust to the use of household or individual-level weights. This is true especially of the cross-country findings for urban and richer households and willingness to get tested for COVID-19: the point estimates tend to be slightly larger when using individual-level compared with household-level weights but they are of comparable magnitude and in most cases retain statistical significance (online supplemental table A7). In contrast, the findings on education have similar point estimates whether we use household or individual-level weights, but with individual-level weights they retain statistical significance only in Malawi. This is likely because individual-level weight recalibration increases the variance of the estimates. All in all, we take this set of tests to indicate the robustness of our results to concerns around respondent selection.

Another potential limitation is the possible malleability of attitudes towards vaccinations, which may have changed, and continue to change, in light of the development of various vaccines and the relative success these appear to have in stemming the pandemic in other parts of the world. Similarly, a thorough investigation into the causal factors underlying country-level and individual-level differences in COVID-19 acceptance rates is beyond the scope of this paper. Future research is thus needed to determine evolving attitudes in Africa towards being vaccinated, their interactions with vaccine supply and availability and the driving forces behind variation in acceptance rates at the country and individual level.

## Conclusions and policy implications

Our headline results of high vaccine acceptance in a cross-country comparable sample of six sub-Saharan African countries suggests that inadequate demand is unlikely to represent the key bottleneck to reaching high COVID-19 vaccine coverage in the region. As willingness to be vaccinated does not automatically translate into vaccine-seeking behaviour, public authorities need to turn intent into effective demand as vaccine rollout progresses.[41] For this, our study identifies some indicative pockets of hesitancy, particularly in Mali, urban Burkina Faso and Malawi, among women in Nigeria, and for richer households and those with more education. Many of these population groups are comparatively easy to reach early in the vaccine rollout process as well as better reachable through targeted communication. Therefore, information, sensitisation, and engagement campaigns that raise acceptance for a COVID-19 vaccine in these clusters and help to maintain the generally high acceptance rates we find will be key. These should focus on resolving concerns about side effects and bolster confidence in the safety of approved COVID-19 vaccines in order to reach mass coverage and end the pandemic swiftly and everywhere.

**Contributors** All authors had full access to all of the data in the study and take responsibility for the integrity of the data and the accuracy of the data analysis. All authors were responsible for its conception and design. YM, PW and SK were responsible for the preparation and analysis of the data and all authors contributed to their interpretation. YM and PW drafted the manuscript, and AZ made critical revision of the manuscript for important intellectual content. SK and AZ conceptualised an early version. AZ is the guarantor. The corresponding author attests that all listed authors meet authorship criteria and that no others meeting criteria have been omitted.

**Funding** Funding for the analysis comes from the World Bank Multi-Donor Trust Fund for Integrated Household and Agricultural Surveys in Low and Middle-Income Countries (TF072496).

**Disclaimer** The funders had no role in study design, data collection, analysis, interpretation of the data, decision to publish or preparation of the manuscript.

**Competing interests** None declared.

**Patient consent for publication** Not applicable.

**Ethics approval** This study involves human participants. Each phone survey was implemented by the respective national statistical office (NSO), except for Ethiopia, where a private firm was the implementing agency. In Burkina Faso, Malawi, Mali, Nigeria and Uganda, the NSO conducts the survey as the sole official statistical authority in the country and in accordance with the respective National Statistical Act, which exempts the NSO from institutional ethics approvals. Informed consent was received from all survey respondents in each country. The World Bank does not require institutional ethics approval for household surveys that are partly or fully financed by the World Bank, including the national phone surveys in Burkina Faso, Ethiopia, Malawi, Mali, Nigeria and Uganda that inform our research.

**Provenance and peer review** Not commissioned; externally peer reviewed.

**Data availability statement** Data are available in a public, open access repository. The data and country-specific questionnaires are available from the World Bank's Microdata Library, High Frequency Phone Survey Catalog (https://microdata.worldbank.org/index.php/catalog/hfps) as well as Living Standards Measurement Study catalog (https://microdata.worldbank.org/index.php/catalog/lsms).

**ORCID iDs**
Yannick Markhof http://orcid.org/0000-0001-5654-9033
Alberto Zezza http://orcid.org/0000-0003-1842-1682

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
