## [Reviewer comments · BMJ Open]

ARTICLE DETAILS

TITLE (PROVISIONAL)	The Acceptance of COVID-19 Vaccines in Sub-Saharan Africa: Evidence from Six National Phone Surveys
AUTHORS	Kanyanda, Shelton; Markhof, Yannick; Wollburg, Philip; Zezza, Alberto

VERSION 1 – REVIEW

REVIEWER	Ching Sin Siau Universiti Kebangsaan Malaysia
REVIEW RETURNED	14-Jul-2021

GENERAL COMMENTS	This is an important study from countries (six Sub-Saharan African countries) where research on COVID-19 vaccine acceptance is sorely needed. The study has many strengths, such as its utilization of phone surveys and nationally representative samples, with the application of weights. Here are some minor recommendations o the authors' consideration: Introduction: The authors' statement that "there is a dearth of large-scale evidence on vaccine acceptance in low income countries in general and SSA in particular" is very true. Nevertheless, I would like to recommend that the authors to conduct a simple review of extant literature on the prevalence of vaccine acceptance in the countries studied. Those may not be nationally representative studies, but still may be worth mentioning to highlight the current situation regarding vaccine acceptance in SSA. I have noticed that the discussion cited a number of studies on acceptance rates. It may be good to reflect some of these in the introduction to provide strength to the strength/rationale of the study. Methods: Reasons for vaccine hesitancy/refusal were mentioned in the methods section in detail, and the results were reflected in Figure 2. Were they entered into the multivariate logit regression? If not, I would recommend doing so, as this would provide more insight into whether the reasons for vaccine hesitancy/refusal were statistically associated with the study participants' vaccine acceptance. The authors described that weights were used. Would it be more relevant if the authors could explain further on what parameters were adjusted? Results: It was mentioned that Mali has a "somewhat lower" level of willingness to accept the vaccine. Is there a statistically between-country difference in the prevalence of vaccine acceptance? If this is not known, I would like to suggest that the authors conduct a one-way ANOVA test, or Kruskal Wallis test (or any other appropriate analysis) to determine this. This would enable the comparison
---

	between the countries to be more robust and then, the authors could state with more confidence which of the countries has "lower" or "higher" acceptance of the vaccine.
--	--

REVIEWER	Joseph Nelson Siewe Fodjo Global Health Institute, University of Antwerp
REVIEW RETURNED	15-Jul-2021

GENERAL COMMENTS	GENERAL COMMENTS  - Well written paper investigating covid vaccination behavior in the Sub-Saharan Africa setting. I particularly commend the efforts of the authors to reduce sampling bias which constitutes a major caveat in most covid-19 online surveys available in the literature. - However, I recommend caution regarding the conclusion that vaccine supply is the main bottleneck to achieve reasonable coverage for covid vaccination in Africa. As the authors rightly state, "willingness to be vaccinated does not automatically translate into vaccine-seeking behavior." The concerns about vaccine safety and the malleability of vaccine attitudes could result in low vaccine coverage even if the problem of vaccine supply is resolved. Therefore I suggest tuning down the conclusion in that light, and prioritizing a message that brings forth possible actions to be done to achieve/maintain high vaccine coverage in the target countries. - I would also recommend that the authors consider the paper by Bono et al (Factors Affecting COVID-19 Vaccine Acceptance: An International Survey among Low and Middle-Income Countries. Vaccines 2021, 9, 515. https://doi.org/10.3390/vaccines9050515) to enrich their discussion. That paper also attempts to investigate covid vaccine acceptance in some African countries via online surveys, and the findings are quite interesting and worth discussing with the current manuscript. - Finally in the reference list, the authors sometimes provide insufficient details about the document being cited. For instance, Reference #14 is tagged "Forthcoming" (unclear if it is in the press somewhere or still to be submitted); References #16, #19, and #34 provide no details of the weblinks or description of sources to retrieve the cited articles. Kindly review.
---

REVIEWER	Caijun Sun Sun Yat-Sen University
REVIEW RETURNED	30-Jul-2021

GENERAL COMMENTS	This manuscript performed a phone survey to investigate the acceptance and vaccine hesitancy of COVID-19 vaccines in 6 countries of Sub-Saharan Africa. There are some interesting data, but there are numerous major and minor concerns regarding this manuscript.  1. 11,338 respondents from 6 countries were included in this survey. Please state how to calculate the minimum sample size of participants in one country. Also state the criteria why selecting these 6 countries to represent the Sub-Saharan Africa? 2. The vaccine acceptance varied in different countries, for example, Ethiopia (97.9%) vs Mali (64.5%), and authors should discuss the potential reasons for this difference. 3. Although the author mentioned that it is important to understand the reasons for the delay or refusal of vaccination (vaccine hesitancy), they did not further analyze the possible correlates and reasons for vaccine hesitancy. In addition, this manuscript mentioned the differences in vaccine acceptance willingness
--

	between men and women, but does not further explore the differences in other factors. 4. Was a pre-investigation performed to validate the correctness of their phone questionnaire? Please state how or why not? Also please attach the questionnaire as an Appendix to help readers better understand the study aims and contents. 5. Page 4, Line 13: Sub-Saharan Africa (SSA), should define the SSA in this line 7. 6. Line 14-15: Authors mentioned "those with more years of education are significantly less willing to be vaccinated", please discuss the possible reason. 7. Please state the sampling method in this study. Why chose this six countries? How to determine how many participants were involved in 8. Throughout the manuscript, covid-19 should be COVID-19; *** $p < 0.01$, ** $p < 0.05$, * $p < 0.1$, should be **** $p < 0.001$, ** $p < 0.05$, * $p < 0.01$... 9. This manuscript does not mention the methods of quality control. The survey was implemented by the local bureaus of different countries. Investigators need to be trained to unify the survey standards, otherwise it is easy to lead to bias. 10. The quality of English writing needs improvement.
--	---

VERSION 1 – AUTHOR RESPONSE

Reviewer: 1

Dr. Ching Sin Siau, Universiti Kebangsaan Malaysia

This is an important study from countries (six Sub-Saharan African countries) where research on COVID-19 vaccine acceptance is sorely needed. The study has many strengths, such as its utilization of phone surveys and nationally representative samples, with the application of weights. Here are some minor recommendations for the authors' consideration:

Comment: Introduction: The authors' statement that "there is a dearth of large-scale evidence on vaccine acceptance in low income countries in general and SSA in particular" is very true. Nevertheless, I would like to recommend that the authors to conduct a simple review of extant literature on the prevalence of vaccine acceptance in the countries studied. Those may not be nationally representative studies, but still may be worth mentioning to highlight the current situation regarding vaccine acceptance in SSA. I have noticed that the discussion cited a number of studies on acceptance rates. It may be good to reflect some of these in the introduction to provide strength to the strength/rationale of the study.

Reply: Thank you for the feedback and for taking the time to review this study. We agree with the importance of citing previous literature on the prevalence of vaccines acceptance to the general framing of this study. In bringing these in the 'Discussion' part of the paper, we are following BMJ Open's style guidelines for research articles and best practices guide. However, following your suggestion we now hint at some of the existing evidence at the end of paragraph 2 of the introduction.

Comment: Methods: Reasons for vaccine hesitancy/refusal were mentioned in the methods section in detail, and the results were reflected in Figure 2. Were they entered into the multivariate logit regression? If not, I would recommend doing so, as this would provide more insight into whether the reasons for vaccine hesitancy/refusal were statistically associated with the study participants' vaccine acceptance.

Reply: Thank you for this comment. In the surveys we rely on, respondents were first asked about their willingness to take a free and safe vaccine against COVID-19. Only respondents who answered 'no' to this vaccine acceptance question were asked what reasons they had for not wanting to take a free and safe vaccine. In Figure 2, we show the responses to that latter question, i.e. the reasons for not wanting to take a free and safe vaccine. Hence for Figure 2 we only use the sample of respondents that stated they do not want a free and safe vaccine. In contrast, in the main multivariate

regression table, we use the full sample to analyze the correlates of acceptance. For this reason, we cannot enter the reasons for refusal in the main multivariate logit regression. What we could do in theory is run a new set of multivariate regressions to analyze the correlates of different reasons for vaccine hesitancy. However, the sample sizes would be very small and we therefore decided against that.

Comment: The authors described that weights were used. Would it be more relevant if the authors could explain further on what parameters were adjusted?

Reply: Thank you for your comment. We agree that this information is important to the present study. To recalibrate the weights, we follow the approach set out in Ambel et al. (2021) (household weights) and Brubaker et al. (2021) (individual-level weights) including the parameters/covariates described therein. The overall strategy employed in these papers as well as by us goes back to the seminal methodological paper by Himelein (2014). Following your feedback, we now mention the types of covariates used in the logistic response propensity model briefly in the text, i.e. household characteristics such as household size, urban/rural residence, economic status, education, sex and age of the household head, and a number of other household characteristics selected based on their observed bias in the interviewed sample (and equivalently individual characteristics for the individual weights such as sex, age, education, marital status, relation to the household head, employment status, and mobile phone ownership).

Comment: Results: It was mentioned that Mali has a "somewhat lower" level of willingness to accept the vaccine. Is there a statistically between-country difference in the prevalence of vaccine acceptance? If this is not known, I would like to suggest that the authors conduct a one-way ANOVA test, or Kruskal Wallis test (or any other appropriate analysis) to determine this. This would enable the comparison between the countries to be more robust and then, the authors could state with more confidence which of the countries has "lower" or "higher" acceptance of the vaccine.

Reply: Thank you for this comment. We have followed your suggestion and added a test for the statistical significance of the difference in acceptance rates between each country pair in Table A.4. Specifically, we perform a set of bivariate regressions for each country pair according to

$$y_{i,j} = \beta_0 + \beta_1 \text{Country}_k + \varepsilon$$

where the sample only ever comprises observations from the respective country pair, y is the dummy for whether individual i in country j accepts the vaccine and β_1 is the coefficient on a dummy for one of the two countries comprising the sample for the regression. As such, β_1 indicates the presence or absence of a statistically significant difference in the mean acceptance rate between the two countries. The new Table A.4 reports p -values for β_1 for each country pair. We felt like this pairwise analysis allowed for more conclusive insights into the significance of cross-country differences in acceptance rates than omnibus tests such as one-way ANOVA or a Kruskal Wallis test that only investigate whether there are any differences across the entire set of countries.

Reviewer: 2

Dr. Joseph Nelson Siewe Fodjo, Global Health Institute, University of Antwerp, Brain Research Africa Initiative (BRAIN)

Comment: GENERAL COMMENTS

- Well written paper investigating covid vaccination behavior in the Sub-Saharan Africa setting. I particularly commend the efforts of the authors to reduce sampling bias which constitutes a major caveat in most covid-19 online surveys available in the literature.

Reply: Thank you very much for your insightful comments on this paper.

Comment: - However, I recommend caution regarding the conclusion that vaccine supply is the main bottleneck to achieve reasonable coverage for covid vaccination in Africa. As the authors rightly state, "willingness to be vaccinated does not automatically translate into vaccine-seeking behavior." The concerns about vaccine safety and the malleability of vaccine attitudes could result in low vaccine coverage even if the problem of vaccine supply is resolved. Therefore I suggest tuning down the conclusion in that light, and prioritizing a message that brings forth possible actions to be done to achieve/maintain high vaccine coverage in the target countries.

Reply: *Thank you, we agree with the points made in this comment and have revised the conclusion accordingly.*

Comment: - I would also recommend that the authors consider the paper by Bono et al (Factors Affecting COVID-19 Vaccine Acceptance: An International Survey among Low and Middle-Income Countries. *Vaccines* 2021, 9, 515.

[https://nam11.safelinks.protection.outlook.com/?url=https%3A%2F%2Fdoi.org%2F10.3390%2Fvaccines9050515&data=04%7C01%7Cymarkhof%40worldbank.org%7Cd0f36bbcfbf844aa88fd08d9536e18b3%7C31a2fec0266b4c67b56e2796d8f59c36%7C0%7C0%7C637632554595820063%7CUnknown%7CTWFpbGZsb3d8eyJWljoimc4wLjAwMDAiLCJQIjoiV2luMzliLCJBTiI6IjEkaWwWLCJXVCi6Mn0%3D%7C1000&data=EdU13PkjOcnNPzuq4Q2BFqwuHb2ZNQz9bv%2B%2fatW1rQ%3D&mp:reserved=0](https://nam11.safelinks.protection.outlook.com/?url=https%3A%2F%2Fdoi.org%2F10.3390%2Fvaccines9050515&data=04%7C01%7Cymarkhof%40worldbank.org%7Cd0f36bbcfbf844aa88fd08d9536e18b3%7C31a2fec0266b4c67b56e2796d8f59c36%7C0%7C0%7C637632554595820063%7CUnknown%7CTWFpbGZsb3d8eyJWljoimc4wLjAwMDAiLCJQIjoiV2luMzliLCJBTiI6IjEkaWwWLCJXVCi6Mn0%3D%7C1000&data=EdU13PkjOcnNPzuq4Q2BFqwuHb2ZNQz9bv%2B%2FatW1rQ%3D&mp:reserved=0) to enrich their discussion. That paper also attempts to investigate covid vaccine acceptance in some African countries via online surveys, and the findings are quite interesting and worth discussing with the current manuscript.

Reply: *Thank you for drawing our attention to this study. We are now citing it at the relevant points in the paper.*

Comment: - Finally in the reference list, the authors sometimes provide insufficient details about the document being cited. For instance, Reference #14 is tagged "Forthcoming" (unclear if it is in the press somewhere or still to be submitted); References #16, #19, and #34 provide no details of the weblinks or description of sources to retrieve the cited articles. Kindly review.

Reply: *Thanks for pointing this out. We have made sure to fix this.*

Reviewer: 3
Dr. Caijun Sun, Sun Yat-Sen University

This manuscript performed a phone survey to investigate the acceptance and vaccine hesitancy of COVID-19 vaccines in 6 countries of Sub-Saharan Africa. There are some interesting data, but there are numerous major and minor concerns regarding this manuscript.

Comments:

1. 11,338 respondents from 6 countries were included in this survey. Please state how to calculate the minimum sample size of participants in one country. Also state the criteria why selecting these 6 countries to represent the Sub-Saharan Africa?

4. Was a pre-investigation performed to validate the correctness of their phone questionnaire? Please state how or why not? Also please attach the questionnaire as an Appendix to help readers better understand the study aims and contents.

7. Please state the sampling method in this study. Why chose this six countries? How to determine how many participants were involved in.

9. This manuscript does not mention the methods of quality control. The survey was implemented by the local bureaus of different countries. Investigators need to be trained to unify the survey standards, otherwise it is easy to lead to bias.

Reply: *Thank you for this series of important comments. We take the liberty to address them jointly as we feel they all relate to the point of insufficient background information on the design and implementation of the surveys and survey instruments in the previous version of our manuscript.*

The High Frequency Phone Surveys (HFPS) from which we take our data were conceived in March 2020 by a working group formed within the World Bank. They are multi-topic, longitudinal surveys consisting of (i) core modules with core questions that are repeated in each survey round, (ii) optional modules for which inclusion was within the discretion of NSOs according to country need and context, (iii) rotating modules with questions that are only periodically featured in specific survey rounds. The questions on vaccine acceptance belong to the latter category of rotating questions and at the time of writing had only been featured once in the fall of 2020.

In order to safeguard cross-country comparability and maintain the same high standard across implementing NSOs, dedicated working groups within the World Bank elaborated standardized guidelines for sampling (accessible here), questionnaire design (guidelines here, questionnaire template here and interviewer manual here), and implementation (here). These guidelines were circulated ahead of the commencement of the surveys and peer-reviewed inside and outside the World Bank. In the following, we pick up the points you raise in particular in your comments above.

Your comments number 1 and 7 refer to sample selection. The countries covered in our study represent all countries that had their HFPS supported by the Living Standards Measurement Study team within the World Bank (see here). These six countries were the ones where we had rich, pre-COVID, and cross-country comparable data from face-to-face household surveys that were used as sampling frames. With regard to the LSMS-supported HFPS, Carletto and Kilic (2020) provide a detailed discussion of their conception and features.

The pre-COVID face-to-face surveys attempted to collect phone numbers from all household members (or alternatively a reference contact such as a neighbor). The sampling for the HFPS started with all enumeration areas covered in the latest pre-COVID face-to-face household survey. Depending on budget availability, either all households with at least one phone number for a household member or reference contact (Ethiopia, Malawi, and Uganda) or a random sub-sample (Burkina Faso, Nigeria, and Mali) was selected for interviewing. Chapter 2.1 of Brubaker et al. (2021) contains a country-by-country discussion of contact protocols. The target sample size for the HFPS was between 1,500 and 1,800 households, a figure that the survey rounds we take our data on vaccine willingness from all meet or exceed (see Appendix Table A.1). The target sample size was selected in order to ensure that, at a minimum, it would be sufficient to detect a 10 percentage point change in the key indicators (COVID-19 knowledge and behavior and labor market impacts) in between rounds with 90% power and 95% confidence at the national level. Note that these sample size requirements for reliable detection of changes across survey rounds are substantially higher than what is required for reliable point estimates within a single round as in the case of our study (for a detailed discussion, you may refer to Chapter 2 of the sampling guidelines).

Your comment number 4 concerns piloting of and access to the full questionnaires. To ensure smooth implementation and common standards across countries, the launch of the first round of HFPS was preceded by three days of piloting the questionnaire, CATI technology, survey protocols, and monitoring mechanisms (for more information, see the CATI implementation guidelines here). Questionnaire guidelines and templates were first elaborated by the World Bank's COVID-19 questionnaire working group and served as the backbone of the HFPS implemented in each NSO (see overview, template, and interview manual). They consisted of (i) a core set of modules with core questions that were repeated in each survey round, (ii) optional modules whose inclusion was within the discretion of the respective NSO, and (iii) rotating modules of which the one on vaccine acceptance was an example. The country-specific questionnaires implemented in the respective survey round from which our data on vaccine acceptance stem are openly available together with the data in the World Bank's Micro Data Library. Following your important comment, we have made sure this is now reflected in the data availability statement of the manuscript so readers can refer to the respective full questionnaires should they wish to.

Lastly, you correctly point out in your comment number 9 that common implementation standards are key to ensure consistent data quality across countries. Interviewers for the survey were selected out of a pool of existing enumerators with experience conducting the LSMS-ISA household surveys. As such, all enumerators had undergone previous LSMS training and were intimately familiar with LSMS-style surveys. In some of the countries, enumerators furthermore had previous experience conducting surveys over the phone. To prepare enumerators specifically for conducting the HFPS, all

interviewers received three days of standardized training ahead of the first round of the survey. Additionally, interviewers received a one-day follow-up training in between survey rounds. Regular audio audits ensured consistently high quality between interviewers. The guideline interview manual shared with enumerators across countries can be accessed here.

In response to your feedback, we have ensured that more information on the conception and implementation of the HFPS is now also reflected in the supplementary annex and referenced in the Methods section of the manuscript.

Comment: 2.The vaccine acceptance varied in different countries, for example, Ethiopia (97.9%) vs Mali (64.5%), and authors should discuss the potential reasons for this difference.

Reply: Thank you for your comment. We agree that explaining cross-country differences in acceptance rates would be a natural next step and highly relevant insight. However, at its core, our results are of descriptive nature with their emphasis on an accurate estimation of country-level acceptance rates rather than causal inference. We feel like a confident exploration of the factors explaining country-level differences would take a more thorough investigation into a small set of specific hypotheses than what is within the scope of our paper. For example, rates may differ due to (i) differences in the average respondent profile across countries and (ii) differences in country-level factors such as politics, the course of the pandemic, and culture. Factors in (i) may cancel each other out and would require high confidence in accurate point estimates for correct inference. Similarly, factors in (ii) are highly complex and would require a dedicated exploration in a large cross-country analysis and/or in-depth case studies of country context. Both unfortunately lie outside the scope of our paper. As any discussion of the potential reasons for the cross-country differences we find would thus be speculative, we fear this could be misleading rather than add to the paper. We therefore prefer keeping the discussion to a predominantly descriptive exposition of cross-country differences in vaccine acceptance rates and the factors associated with higher hesitancy. We have taken care to make this limitation more explicit in the respective section of the Discussion.

Comment: 3. Although the author mentioned that it is important to understand the reasons for the delay or refusal of vaccination (vaccine hesitancy), they did not further analyze the possible correlates and reasons for vaccine hesitancy. In addition, this manuscript mentioned the differences in vaccine acceptance willingness between men and women, but does not further explore the differences in other factors.

Reply: Thank you for your comment. In the Results chapter of our manuscript under the “Correlates of hesitancy” sub-heading, we provide multivariate logit regressions of vaccine acceptance on a number of individual and household characteristics for each country. As such, these regressions investigate the factors associated with vaccine acceptance (or hesitancy) for a set of characteristics including, but not limited to, the factors by which we disaggregate vaccine acceptance rates under the “Descriptive results” heading. Furthermore, we explore the robustness of these results to the use of individual-level survey weights in the Discussion section and Table A.7. This analysis provides insights into differences in vaccine hesitancy between respondents such as men and women but also depending on a large set of additional individual and household characteristics. Please note that we refrain from giving them a causal interpretation throughout our manuscript as this would require a more thorough investigation of a small set of specific hypotheses than what is within our paper’s scope.

Comment: 5.Page 4, Line 13: Sub-Saharan Africa (SSA) , should define the SSA in this line 7.

Reply: Thanks for pointing this out, we have adjusted it.

Comment: 6.Line14-15: Authors mentioned “those with more years of education are significantly less willing to be vaccinated”, please discuss the possible reason.

Reply: Thank you for your comment. Together with our findings of higher hesitancy in urban areas and among richer households, the finding that those with more education are less willing to be vaccinated may relate to differences in the main information source, particularly online sources/social media, between respondents. Use of online sources/social media has been linked to having been subjected to misinformation about COVID-19, holding conspiracy beliefs concerning the disease, and higher hesitancy toward a COVID-19 vaccine (e.g., Romer and Jamieson (2021), Africa CDC (2021)). In the context of the countries comprising our sample, the use of such online sources is likely more common among respondents with higher socio-economic status and areas with better infrastructure, possibly explaining the effects we find with regard to the above covariates (including years of education). Based on our data, such discussion is speculative, however, and causal identification would take a more thorough investigation into this hypothesis than what is within the scope of this paper. We thus refrain from providing this kind of causal discussion in the manuscript (see also our reply to comments 2 and 3). However, we have added some additional pointers to acknowledge this important limitation more clearly.

Comment: 8. Throughout the manuscript, covid-19 should be COVID-19; *** p<0.01, ** p<0.05, * p<0.1, should be “*** p<0.001, ** p<0.05, * p<0.01”...

Reply: Thank you for bringing this more conventional spelling to our attention, we have adjusted it accordingly throughout the manuscript. Thank you also for suggesting the use of alternative thresholds to denote significance in our regression tables. We are not aware of an official editorial guideline for BMJ Open regarding this, however, we have chosen the threshold levels of *** p<0.01, ** p<0.05, and * p<0.1 for the asterisks in the regression tables in compliance with the standard for social science research. In addition, we provide robust standard errors together with our point estimates to be transparent about the exact level of significance for the interested reader. Should the standard BMJ Open policy deviate from our current presentation though and editors prefer amendment, we are of course happy to make the requisite changes.

Comment: 10. The quality of English writing needs improvement.

Reply: The manuscript has now undergone a round of language editing from a professional editor between the initial submitted manuscript and the resubmission.

VERSION 2 – REVIEW

REVIEWER	Ching Sin Siau Universiti Kebangsaan Malaysia
REVIEW RETURNED	24-Aug-2021
GENERAL COMMENTS	The reviewer completed the checklist but made no further comments.
REVIEWER	Joseph Nelson Siewe Fodjo Global Health Institute, University of Antwerp
REVIEW RETURNED	24-Aug-2021
GENERAL COMMENTS	Thank you for the corrections made. I have no further comments.
REVIEWER	Caijun Sun Sun Yat-Sen University
REVIEW RETURNED	03-Sep-2021
GENERAL COMMENTS	Authors have correctly addressed my concerns.